# Reference gene expression stability within the rat brain under mild intermittent ketosis induced by supplementation with medium-chain triglycerides

**Alexander P. Schwarz**[1]☯*, **Veronika A. Nikitina**[2]☯, **Darya U. Krytskaya**[2], **Ksenia P. Shcherbakova**[2], **Alexander N. Trofimov**[2]*

1 Laboratory of Molecular Mechanisms of Neuronal Interactions, I.M. Sechenov Institute of Evolutionary Physiology and Biochemistry, Russian Academy of Sciences, St. Petersburg, Russia, 2 Laboratory of Neurobiology of the Brain Integrative Functions, I.P. Pavlov Department of Physiology, Institute of Experimental Medicine, St. Petersburg, Russia

☯ These authors contributed equally to this work.
* aleksandr.pavlovich.schwarz@gmail.com (APS); alexander.n.trofimov@gmail.com (ANT)

**Data Availability Statement:** The data presented in this study are available in the article and its Supporting Information files.

## Abstract

Reverse transcription followed by quantitative (real-time) polymerase chain reaction (RT-qPCR) has become the gold standard in mRNA expression analysis. However, it requires an accurate choice of reference genes for adequate normalization. The aim of this study was to validate the reference genes for qPCR experiments in the brain of rats in the model of mild ketosis established through supplementation with medium-chain triglycerides (MCT) and inter-mittent fasting. This approach allows to reproduce certain neuroprotective effects of the classical ketogenic diet while avoiding its adverse effects. Ketogenic treatment targets multiple metabolic pathways, which may affect the reference gene expression. The standard chow of adult Wistar rats was supplemented with MCT (2 ml/kg orogastrically, during 6 h of fasting) or water (equivolume) for 1 month. The mRNA expression of 9 housekeeping genes (*Actb*, *B2m*, *Gapdh*, *Hprt1*, *Pgk1*, *Ppia*, *Rpl13a*, *Sdha*, *Ywhaz*) in the medial prefrontal cortex, dorsal and ventral hippocampus was measured by RT-qPCR. Using the RefFinder® online tool, we have found that the reference gene stability ranking strongly depended on the analyzed brain region. The most stably expressed reference genes were found to be *Ppia*, *Actb*, and *Rpl13a* in the medial prefrontal cortex; *Rpl13a*, *Ywhaz*, and *Pgk1* in the dorsal hippocampus; *Ywhaz*, *Sdha*, and *Ppia* in the ventral hippocampus. The *B2m* was identified as an invalid reference gene in the ventral hippocampus, while *Sdha*, *Actb*, and *Gapdh* were unstable in the dorsal hippocampus. The stabilities of the examined reference genes were lower in the dorsal hippocampus compared to the ventral hippocampus and the medial prefrontal cortex. When normalized to the three most stably expressed reference genes, the *Gapdh* mRNA was upregulated, while the *Sdha* mRNA was downregulated in the medial prefrontal cortex of MCT-fed animals. Thus, the expression stability of reference genes strongly depends on the examined brain regions. The dorsal and ventral hippocampal areas differ in reference genes stability rankings, which should be taken into account in the RT-qPCR experimental design.

**Funding:** This research was funded by the Russian Science Foundation (RSF, project no. 19-75-10076). The funders had no role in study design, data collection and analysis, decision to publish, or preparation of the manuscript.

## Introduction

Reverse transcription followed by quantitative (real-time) polymerase chain reaction (RT-qPCR) is known to be a gold standard in low throughput mRNA expression analysis. However, it requires an accurate choice of reference genes for normalization to obtain reliable results. Recent findings show that reference gene stability may be affected by the experimental design [1]. The choice of an unstable reference gene for RT-qPCR data normalization could conceal changes of target mRNA levels or erroneously reveal irrelevant effects of experimental treatments [2–8]. The MIQE (minimum information for publication of quantitative real-time PCR experiments) guidelines require experimental determination of the choice of optimal reference genes with clear validation of their stability within the reported experimental conditions [9].

We have recently developed a set of multiplex qPCR assays allowing fast expression of nine housekeeping genes in just 3 reactions [10]. *Gapdh*, *Actb*, *Ppia*, *B2m*, *Hprt1*, *Ywhaz*, *Rpl13a*, *Pgk1*, *Sdha* genes selected for analysis are widely used as reference in RT-qPCR experiments in brain tissues or brain-derived cell lines in laboratory rats and have shown higher or lower stability depending on experimental settings [1]. These genes participate in different cell functions (glucose and energy metabolism, protein folding, protein synthesis, purine synthesis, signaling, antigen presentation, cytoskeleton building; see S1 Table) to minimize possible co-regulation effects, which could impact the analysis results.

In the present work, we aim to evaluate the expression stability of 9 housekeeping genes widely used as reference and select suitable reference genes for RT-qPCR analysis within the prefrontal cortex and hippocampus in the rat model of mild ketosis induced by medium-chain triglyceride (MCT) supplementation. MCT supplementation is considered a promising approach to mimic the neuroprotective effects of the classic ketogenic diet without its adverse effects [11, 12] and has been shown to affect the expression of neuroplasticity-related genes [13] and exert effects in the brain in a region-specific manner [14].

## Materials and methods

### Animals and study design

The study was performed on 9 m.o. male Wistar rats weighed 444 ± 69 g (M ± SD). Animal experiments were carried out under the Guidelines on the Treatment of Laboratory Animals effective at the Institute of Experimental Medicine (St. Petersburg, Russia), and these guidelines comply with EU Directive 2010/63/EU for animal experiments. Experimental protocol was approved by the Ethical committee of the Institute of Experimental Medicine (protocol no. 2/22).

Experimental animals were housed under standard conditions with 12 h light-dark cycle (lights on at 5 AM) and *ad libitum* access to water. The study design is summarized in Fig 1.

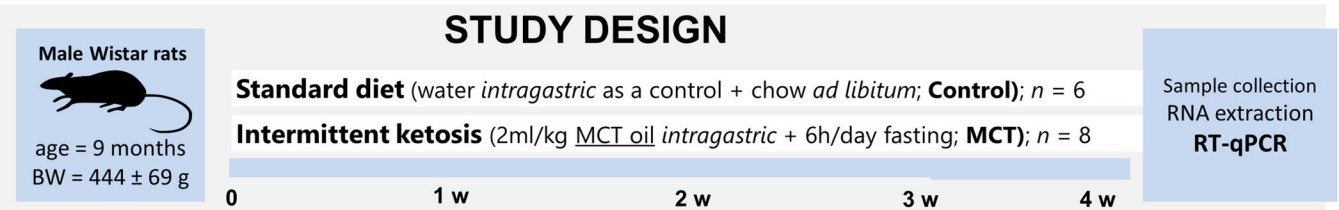

**Fig 1. The design of the study aimed to assess the effect of mild ketosis on the reference gene expression stability in the rat brain.** MCT (2 ml/kg body weight) were given as a supplement to standard feed to adult (9 m.o., BW = 444 ± 69 g) animals by oral gavage daily for 4 weeks, while the control rats received equivolume water. mRNA expression of nine housekeeping genes (*Actb*, *Gapdh*, *B2m*, *Rpl13a*, *Sdha*, *Ppia*, *Hprt1*, *Pgk1*, *Ywhaz*) in the medial prefrontal cortex, dorsal and hippocampal areas was measured by RT-qPCR.

Rats were given MCT (Jarrow Formulas, Inc., Los Angeles CA, USA) at a dose of 2 ml/kg daily by oral gavage during a 6-hour fasting period or an equivolume of water without fasting (control animals) for 4 weeks. Adjusted for metabolic rate [15], this dosage corresponds to approximately 20 g/day of MCT for an adult human subject, a typical dose used in human studies [16]. Animals were sacrificed by decapitation. Brains were quickly collected, frozen, and stored at -70˚C for further analysis.

## Extraction of brain structures and total RNA isolation

Brains were placed into freezing microtome Thermo-scientific™ Microm HM525 (Thermo Fisher Scientific, Waltham, MA, USA) for 60 min at -20˚C and then cut in coronal direction. Medial prefrontal cortex, dorsal and ventral hippocampal regions were captured by microspatula, the structures determined based on the rat brain atlas [17], as described earlier [18]. Samples were homogenized in an appropriate volume of ExtractRNA reagent (Evrogen JSC, Moscow, Russia) and total RNA was extracted by single-step acid guanidinium-thiocyanate-phenol-chloroform method [19] according to the manufacturer's instructions.

## RNA purification

To purify the isolated RNA from possible genomic DNA contamination, the samples were treated with ribonuclease-free deoxyribonuclease (RQ1 DNase; Promega Corp., Madison, WI, USA). The tubes with the previously precipitated RNA were centrifuged for 30 min at +4˚C, 12,000 g, and the ethanol supernatant was removed. The pellet was dried using a dry-block thermostat TS-100 (Biosan, Riga, Latvia) for 5 min at +50˚C. The dried pellet was resuspended in 7.5 μl of milli-Q water and incubated at +50˚C for 5 min for better dissolution of nucleic acids. A reaction mixture containing 10 U of a recombinant RNasin Ribonuclease Inhibitor (Sileks, Moscow, Russia) and 2 U of RQ1 DNase in a 2× reaction buffer was added to the resulting solution. The tubes with the reaction mixture were incubated for 10 min at +37˚C. After incubation, 1.5 μL (1/10 of the volume of the reaction mixture) of a stop reagent (20 mM EGTA) was added to the samples, mixed, incubated at +70˚C to inactivate the enzyme for 5 min, followed by cooling to +4˚C.

To purify the resulting preparation from the reaction components, RNA was precipitated: the volume of the solution was brought to 30 μL with milli-Q water, 3 μL (1/10 of the volume of RNA solution) of 3M sodium acetate solution, and 75 μL (2.5 volumes of RNA solution) of 96% ethanol, thoroughly mixed, and then incubated at –20˚C overnight. After that, the tubes were centrifuged, the precipitate was washed with 75% ethanol and stored under alcohol until further handling.

## RT-qPCR

Before carrying out the reverse transcription (RT) reaction, the tubes with RNA were centrifuged for 30 min at +4˚C, 12,000 g, the RNA sediment was dried and then dissolved in 15 μL of milli-Q water. RNA concentration (determined by measuring the absorbance at 260 nm) and purity (determined by measuring the 260/230 and 260/280 nm absorbance ratios) were measured spectrophotometrically using a Nanodrop 2000 instrument (Thermo Fisher Scientific, Waltham, MA, USA). To carry out the RT reaction, 0.5 μg oligo(dT) and 0.25 μg random-9-mer primers (BioBeagle, St. Petersburg, Russia) were added to 8 μL of the RNA solution containing 1 μg of RNA preliminarily equalized in terms of concentration and incubated at + 70˚C for 5 min. Samples were briefly cooled and centrifuged. Then a reaction mixture containing M-MLV reverse transcriptase (Promega Corp., Madison, WI, USA), a mixture of dNTPs (25 mM dATP, 25 mM dCTP, 25 mM dGTP, 25 mM dTTP; Medigen, Novosibirsk,

Russia), 5× M-MLV reaction buffer (250 mM Tris-HCl, 375 mM KCl, 15 mM MgCl$_2$, 50 mM DL-dithiothreitol, pH 8.3; Promega Corp., Madison, WI, USA), RNasin in deionized water was added. The samples were thoroughly vortexed and incubated for 2 h at +42˚C for the synthesis of cDNA on the RNA template and for 10 min at +65˚C to stop the reaction through enzyme inactivation. The resulting cDNA solution (20 µL) was diluted 5-fold with deionized water and stored at −20˚C until real-time polymerase chain reaction (qPCR) with fluorescent probes (TaqMan technology).

In the present work, we tested a panel of 9 reference genes that regulate different cell functions and are frequently used for RT-qPCR data normalization in the rat brain [1]. We used three triplex qPCR assays validated in our previous work [10]: *Actb+Gapdh+B2m*; *Rpl13a +Sdha+Ppia*; *Hprt1+Pgk1+Ywhaz*. The descriptions of the used primer/probes are summarized in S1 Table.

The Multiplex qPCR reactions had been optimized and fully described in our previous study [10]. Briefly, the reaction mix contained 1 µL of cDNA sample, 0.75 U of TaqM-polymerase (Alkor Bio Group, St. Petersburg, Russia), 200 nM of specific forward and reverse primers, and either 200nM (for *Actb*) or 100 nM (for the rest of the genes) TaqMan probes, 3.5 mM MgCl$_2$, 250 µM dATP/dTTP/dCTP/dGTP in 10 µL total volume of 1× TaqM-reaction buffer. All oligonucleotides were synthesized by DNA-Synthesis Ltd. (Moscow, Russia). All reactions were duplicated with no reverse transcription and no template control. The reactions were run on C1000 Touch Thermal Cycler combined with CFX96 Real-Time detection system (Bio-Rad Laboratories, Inc., Hercules, CA, USA). The thermal settings were as follows: 1 cycle at 95˚C for 15 min (as recommended by the enzyme manufacturer), 5 cycles (without plate read) with a denaturation step (95˚C for 5 s) and an annealing/elongation step (60˚C for 10 s) and then 35 cycles with a denaturation step (95˚C for 5 s) and an annealing/elongation step (60˚C for 10 s) followed by fluorescence plate read (about 13 s). In a separate experiment, we tested the efficiencies of used assays with the same batch of reagents as the main experiment using serial dilution method. All assays show optimal efficiency (96.5%– 109.3%) and R$^2$ (S4–S6 Figs).

## Gene stability analysis

The PCR curves were analyzed with the CFX Manager software (Bio-Rad Laboratories, Inc., Hercules, CA, USA): the quantification cycle (Cq) values were determined by setting a single threshold. We then imported the raw mean Cq data to RefFinder® online tool (https://www. heartcure.com.au/reffinder/) to evaluate the expression stability of the examined genes. RefFinder® utilizes four commonly used algorithms for reference gene expression stability analysis (the comparative Delta-Ct method, BestKeeper, NormFinder, and GeNorm) and calculates comprehensive ranking by geometric averaging of obtained ranks [20, 21]. Briefly, (a) in the analysis by the comparative Delta-Ct method, selection of the most stable reference genes is achieved by comparing the relative expression of pairs of genes in each sample: genes with the smallest mean standard deviation of delta-Ct are considered the most stable [22]; (b) the BestKeeper tool selects the most stable genes by ranking the geometric means of raw Cq values of each gene, comparing them in pairs, and calculating the coefficient of variation, the standard deviation of the Cq values (the lower, the more stable), and the correlation coefficient (the higher, the more stable) [23]; (c) NormFinder's principle of operation is based on a mathematical model that calculates the stability value of candidate reference genes by estimating both the total variation of the reference candidate genes and the variation between subgroups of samples in a sample set [24]; (d) the basic principle behind the GeNorm algorithm is to calculate the stability value M (the lower, the more stable) by averaging the pairwise variations among all candidate genes in all samples [25]. As we have used only assays with optimal

efficiencies (S4–S6 Figs), the data obtained by RefFinder should not be affected by biases in efficiency [26].

### Gene expression analysis and statistics

The mRNA expression levels were assessed by $2^{-\Delta\Delta Ct}$ method [27] using geometric mean of Cqs for the three most stable reference genes (based on ReFinder results) for normalization of the expression data [25]. Data were analyzed in GraphPad Prism, ver. 9. mRNA expression data was Log2-transformed for normalization. Shapiro-Wilk's test was applied to evaluate normality of data distribution. Levene's test was used to evaluate homogeneity of variances between groups. Student's t-test (equal variance) or Welch corrected t-test (unequal variance) were used to compare group means. Statistical significance was set at $p < 0.05$.

## Results

The expression stability of nine reference genes (*Actb*, *B2m*, *Gapdh*, *Hprt1*, *Pgk1*, *Ppia*, *Rpl13a*, *Sdha*, *Ywhaz*) in several brain regions (medial prefrontal cortex, dorsal and ventral hippocampus) of rats in the model of mild ketosis was analyzed by RefFinder® online tool incorporating comparative Delta-Ct [22], BestKeeper [23], NormFinder [24], and GeNorm [25] algorithms. Stability values and rankings obtained by different methods are summarized in Figs 2 and 3, respectively. Detailed results are provided in S1–S3 Figs. As can be seen in Fig 2, the stability values among the examined brain areas are proportionally quite similar regardless of the algorithm chosen for the analysis. Overall, the examined reference genes demonstrated the highest stability in the medial prefrontal cortex and the lowest stability in the dorsal hippocampus (Fig 2).

### Medial prefrontal cortex

*Actb*, *Rpl13a*, and *Ywhaz* were found to be the most stable reference genes within the medial prefrontal cortex of rats subjected to mild ketosis, based on the RefFinder comprehensive ranking (Fig 3A). The *B2m* expression was the least stable. GeNorm identified *Actb* and *Gapdh* as the most stably expressed reference genes, while *B2m* and *Sdha* were identified as the least stable. GeNorm M-values for all genes were below 0.5 cut-off value. NormFinder found *Ppia* and *Rpl13a* to be the most stable, whereas *B2m* and *Sdha* were the least stable. According to the comparative Delta-Ct method, *Rpl13a*, *Ppia*, and *Actb* were the most stable, while *B2m* and *Sdha* were the least stable. BestKeeper identified *Pgk1*, *Sdha*, and *Ywhaz* as the most stably expressed and *B2m* as the least stably expressed.

### Dorsal hippocampus

RefFinder comprehensive ranking identified *Rpl13a*, *Ywhaz*, *Pgk1* as the most stable reference genes in the dorsal hippocampus, whereas *Sdha* expression was the least stable (Fig 3B). *Rpl13a* and *Ywhaz* were found to be the most stable according to GeNorm analysis, and *Gapdh*, *Actb*, and *Sdha* were identified as unstable and invalid reference genes (0.5 cut-off M-value). *Rpl13a* and *Ywhaz* were also found to be the most stable reference genes by NormFinder and comparative Delta-Ct algorithms, whereas *Actb* and *Sdha* expression was the least stable based on these methods. BestKeeper identified *Pgk1* and *Rpl13a* as the most stable and *B2m* as the least stable.

### Ventral hippocampus

*Ywhaz*, *Sdha*, and *Ppia* were most stably expressed in the ventral hippocampus, whereas *Gapdh* and *B2m* were the most unstable, based on RefFinder comprehensive ranking (Fig 3C).

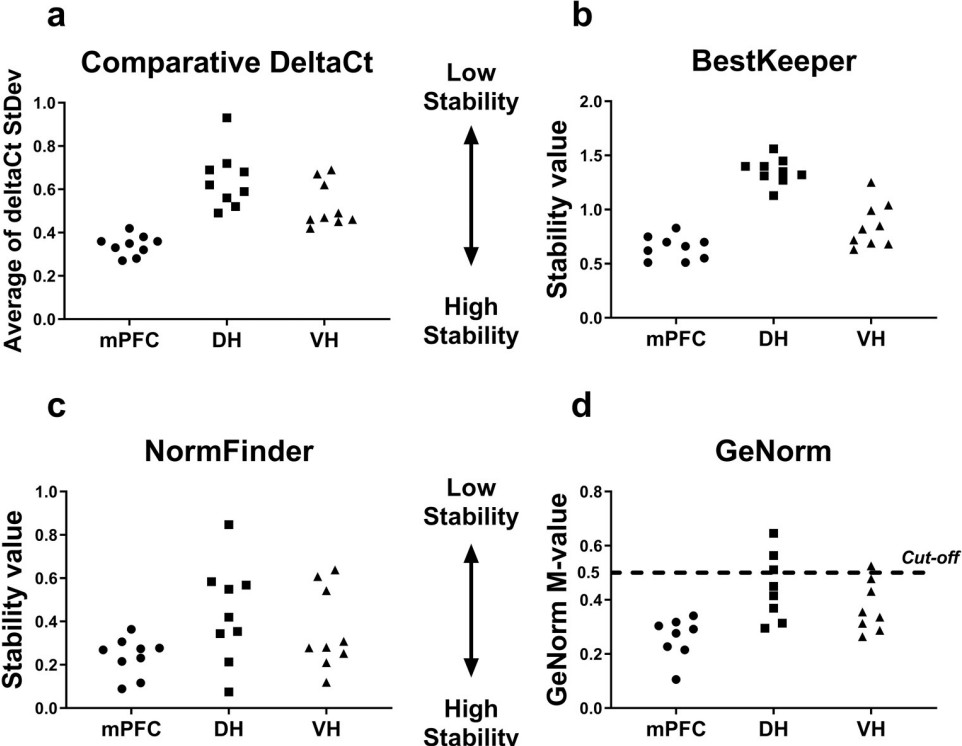

**Fig 2. Reference gene stability values of each housekeeping gene obtained by four methods in three different brain regions of rats fed with MCT.** MCT (2 ml/kg body weight) were given as a supplement to standard feed to adult (9 m. o., BW = 444 ± 69 g) animals by oral gavage daily for 4 weeks, while the control rats received equivolume water. Expression stability values of nine housekeeping genes (*Actb*, *Gapdh*, *B2m*, *Rpl13a*, *Sdha*, *Ppia*, *Hprt1*, *Pgk1*, *Ywhaz*, each represented by a dot per each brain structure) in the medial prefrontal cortex (mPFC, circular dots), dorsal (DH, square dots) and ventral (VH, triangular dots) hippocampal areas were analyzed using **(a)** comparative Delta-Ct, **(b)** BestKeeper, **(c)** NormFinder, and **(d)** GeNorm methods. For GeNorm (D), values above 0.5 cut-off M-value indicate unstable genes inappropriate for RT-qPCR data normalization. Only eight M-values are presented because the GeNorm method calculates a single M-value for the pair of the most stable genes.

*Ywhaz*, *Sdha*, and *Rpl13a* were found to be the most stable, while *Gapdh* and *B2m* were found to be the least stable reference genes according to NormFinder and comparative Delta-Ct results. Best Keeper ranked *Actb*, *Pgk1*, and *Sdha* as the most stable and *B2m* as the least stable reference gene. *Ppia* and *Hprt1* were the best choice reference genes according to the GeNorm algorithm. *B2m* expression was unstable and, therefore, would be invalid for RT-qPCR data normalization according to the GeNorm 0.5 cut-off M-Value.

## The relative mRNA expression of reference genes normalized to the expression of three optimal reference genes

We analyzed the effects of mild ketosis induced by intermittent fasting combined with MCT supplementation on the expression of the analyzed genes normalized to the three most stable reference genes. Within the medial prefrontal cortex, this mild ketogenic intervention downregulated *Sdha* mRNA expression (t = 2.84, df = 12, $p$ = 0.015; Fig 4B), and upregulated *Gapdh* mRNA expression (t = 3.67, df = 12, $p$ = 0.003; Fig 4E). No significant changes were detected in either of the analyzed hippocampal areas (Figs 5 and 6). It should be noted that the variance of *Gapdh* mRNA expression in the ventral hippocampus was higher in MCT-fed animals compared to controls (F = 16.57, $p$ = 0.007).

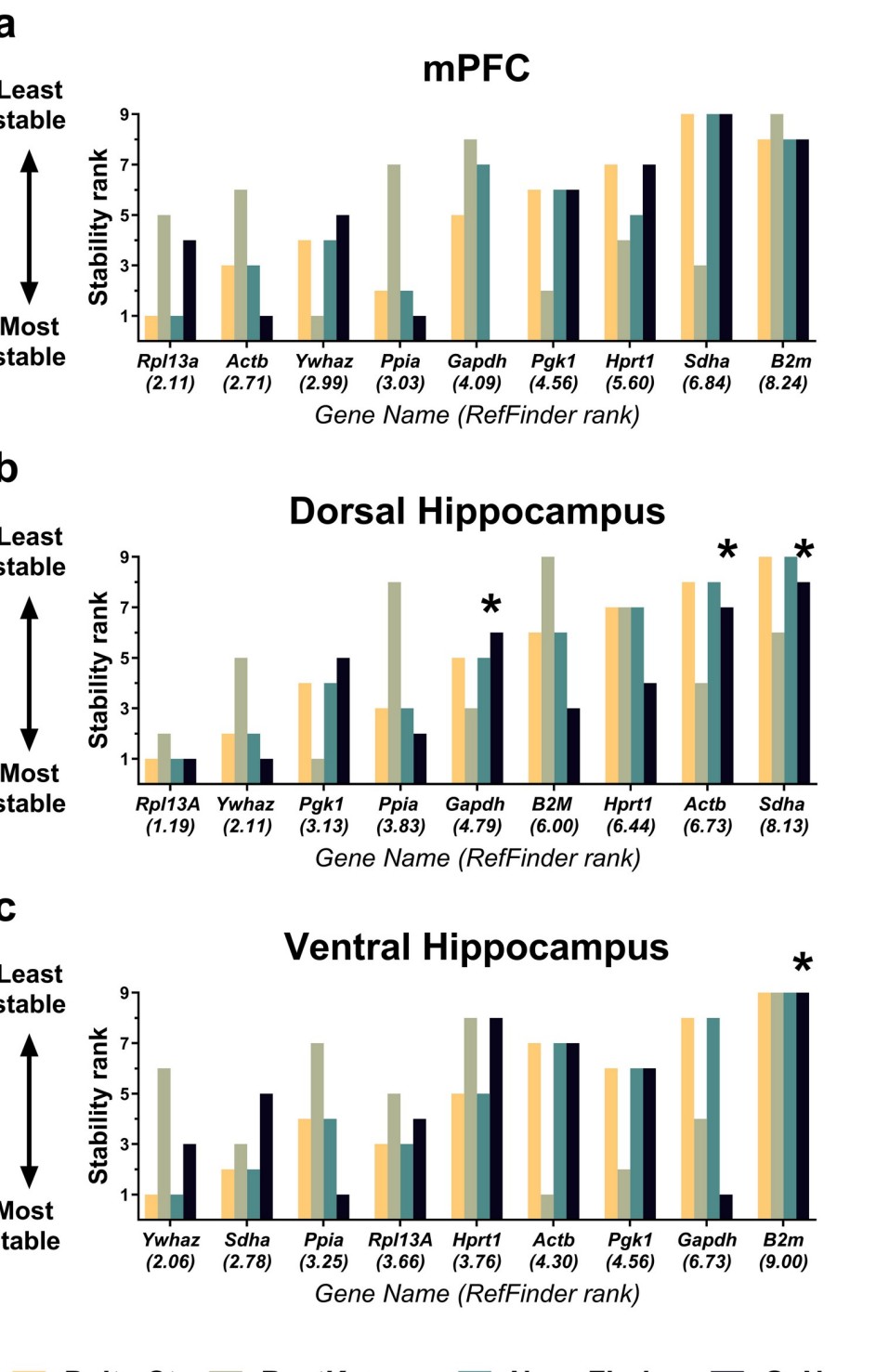

**Fig 3. Reference gene stability rankings in three brain regions of rats fed with MCT.** MCT (2 ml/kg body weight) were given as a supplement to standard feed to adult animals (9 m.o., BW = 444 ± 69 g) by oral gavage daily for 4 weeks, while control rats received equivolume water. Reference gene stability in the **(a)** medial prefrontal cortex, **(b)** dorsal and **(c)** ventral hippocampal areas was analyzed by RefFinder® online tool. RefFinder comprehensive ranking (ranks are indicated in brackets under gene name on X-axis) is based on geometric mean of ranks obtained by four methods: comparative Delta-Ct, BestKeeper, NormFinder, and GeNorm. *–unstably expressed genes inappropriate for normalization, according to the GeNorm cut-off 0.5 M-Value.

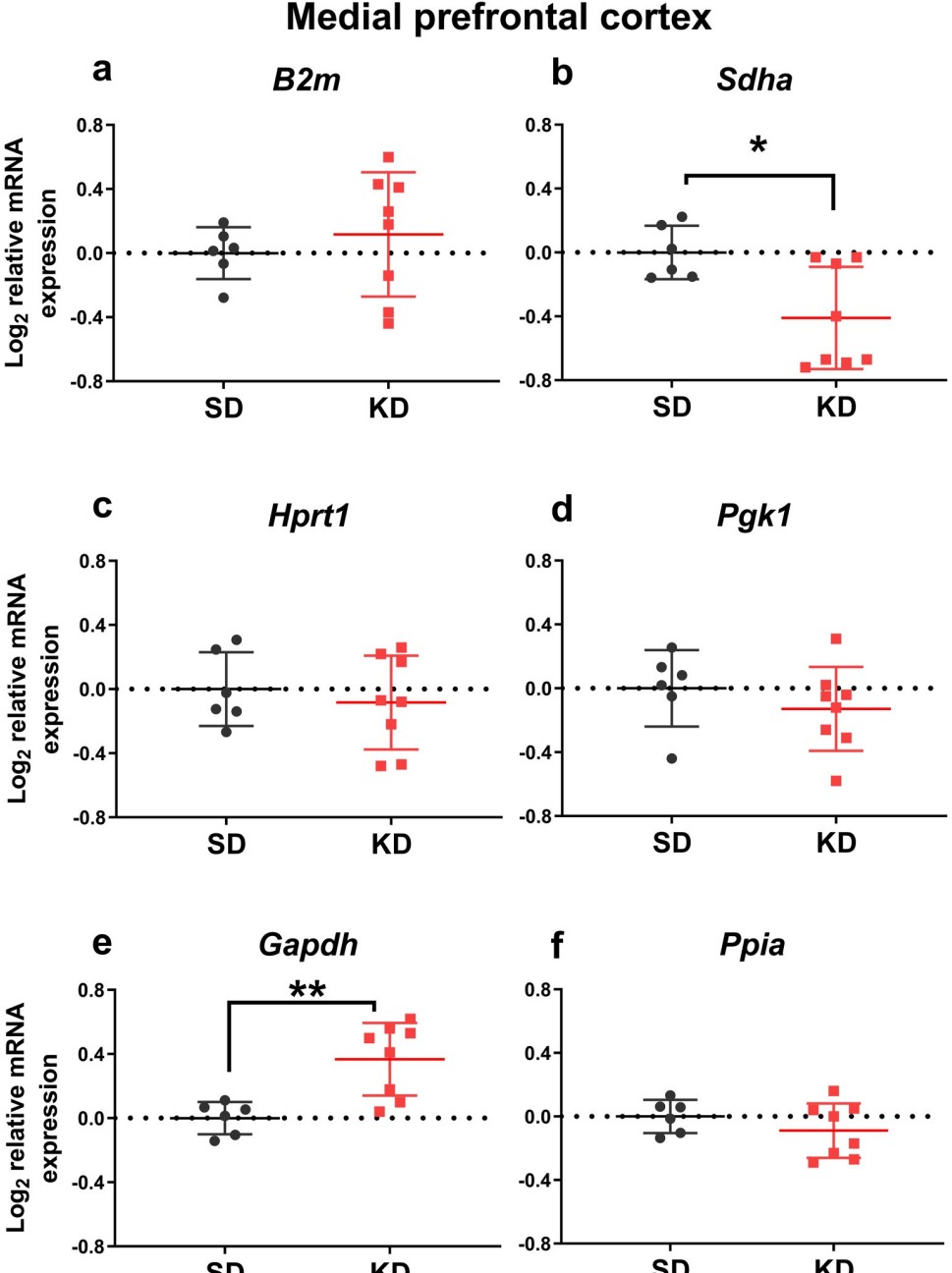

**Fig 4. The effects of medium-chain triglyceride supplementation for 1 month on the mRNA expression of reference genes in the rat medial prefrontal cortex.** MCT (2 ml/kg body weight) were given as a supplement to standard feed to adult animals (9 m.o., BW = 444 ± 69 g) by oral gavage daily for 4 weeks (KD), while the control rats received equivolume water (SD). The mRNA expression was normalized to the geometric mean of the expression level of the three most stably expressed genes (*Rpl13a*, *Actb*, *Ywhaz*). *$-p < 0.05$, **$-p < 0.01$ (Student t-test).

## Discussion

In the present work, we have analyzed the expression stability of nine housekeeping genes within the rat brain in the model of mild ketosis induced by MCT supplementation and intermittent fasting. We found that stability rankings strongly depended on the analyzed brain region.

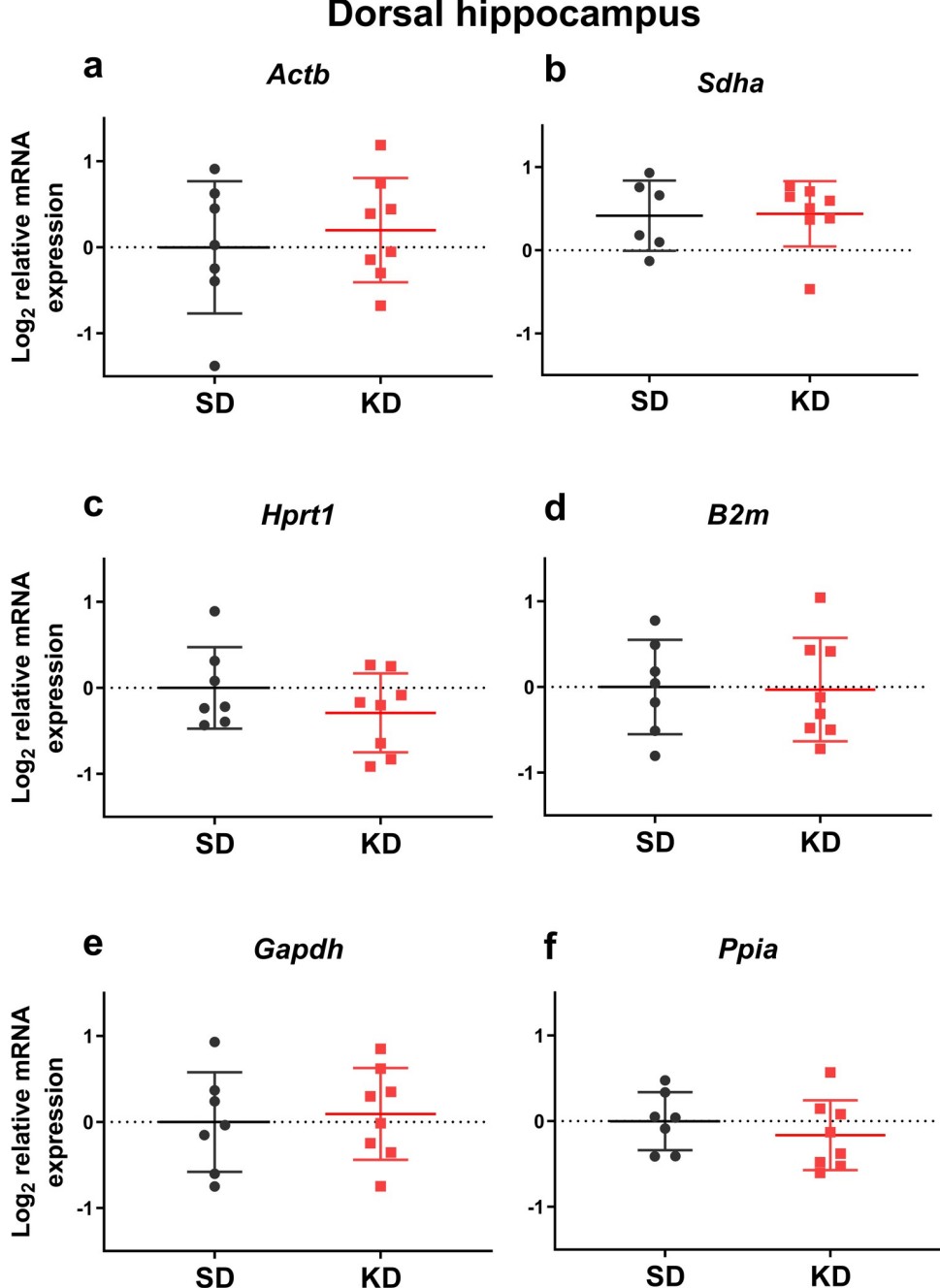

**Fig 5. The effects of medium-chain triglyceride supplementation for 1 month on the mRNA expression of reference genes in the rat dorsal hippocampal area.** MCT (2 ml/kg body weight) were given as a supplement to standard feed to adult animals (9 m.o., BW = 444 ± 69 g) by oral gavage daily for 4 weeks (KD), while the control rats received equivolume water (SD). The mRNA expression was normalized to the geometric mean of the expression level of the three most stably expressed genes (*Rpl13a*, *Ywhaz*, *Pgk1*).

The reference gene stability rankings significantly differed in the dorsal and ventral hippocampal areas. For example, *Sdha* expression was one of the most stable in the ventral hippocampus, while in the dorsal hippocampus, it had low stability and is, therefore, inappropriate for RT-qPCR data normalization. It is well known that dorsal and ventral hippocampal areas

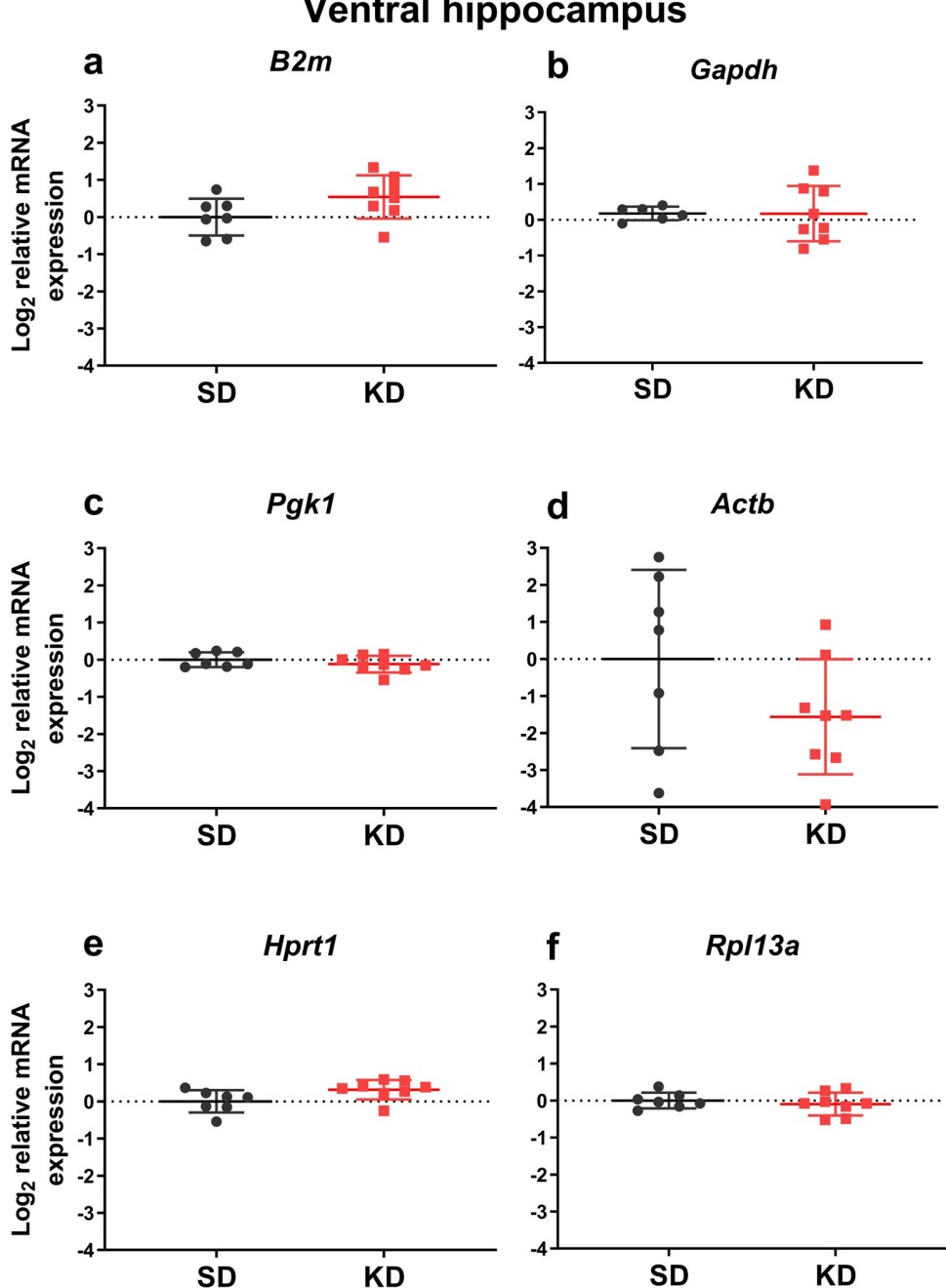

**Fig 6. The effects of medium-chain triglyceride supplementation for 1 month on the mRNA expression of reference genes in the rat ventral hippocampal area.** MCT (2 ml/kg body weight) were given as a supplement to standard feed to adult animals (9 m.o., BW = 444 ± 69 g) by oral gavage daily for 4 weeks (KD), while the control rats received equivolume water (SD). The mRNA expression was normalized to the geometric mean of the expression level of the three most stably expressed genes (*Ywhaz*, *Sdha*, *Ppia*).

have different transcriptomic and functional properties [28]. Our present work, as well as our recent study in the model of juvenile seizures [18], suggest that reference gene stability varies in different hippocampal areas, which should be taken into account during RT-qPCR experimental design. Our findings on the heterogeneity of housekeeping gene expression in the

hippocampus also emphasize that dorsal and ventral parts of the hippocampus should be analyzed separately in biochemical experiments.

The examined reference gene panel, overall, demonstrated lower stability in the dorsal hippocampus compared to the ventral hippocampus and the medial prefrontal cortex (Fig 2). This may reflect a higher sensitivity of the dorsal hippocampus to ketogenic treatment. Our data are in agreement with the evidence that ketogenic diet and MCT supplementation improve dorsal hippocampus-dependent cognitive functions [13, 29]. It should be noted that the protein products encoded by two of the three housekeeping genes with unstable expression in the dorsal hippocampus participate in glucose (glyceraldehyde-3-phosphate dehydrogenase) and energy (succinate dehydrogenase subunit A) metabolism, which are known to be affected by ketogenic diet [29], yet *Sdha* had one of the highest stabilities in the ventral hippocampus. In our previous study in the model of acute seizures, the expression stability of these two genes (*Gapdh* and *Sdha*) in the rat brain varied independently among the analyzed regions [18]. The ketogenic intervention shifted the *Gapdh* and *Sdha* mRNA expression within the medial prefrontal cortex in opposite directions, when normalized to the three optimal reference genes. These changes may reflect region-specific metabolic switching in the rat brain. However, the extent of these changes was relatively small (less than 2-fold) for reliable detection by RT-qPCR analysis.

To sum up, we can conclude that reference gene expression stability varies among brain areas. The housekeeping gene expression seems to be affected by MCT supplementation in the dorsal hippocampus to a greater extent than in the ventral hippocampus and medial prefrontal cortex.

## Supporting information

**S1 Table. Primers and probes used for RT-PCR.**
(XLSX)

**S1 Fig. Stability values obtained by RefFinder online tool in the medial prefrontal cortex (mPFC) of rats fed with medium-chain triglycerides (MCT).** MCT oil (2 ml/kg body weight) was given as a supplement to standard feed to adult (9 m.o., BW = 444 ± 69 g) animals by oral gavage daily for 4 weeks, while the control rats received equivolume water. (a) Coronal rat brain atlas diagrams (reproduced from Paxinos and Watson, 2007) showing mPFC (pink) boundaries at Bregma 4.20 mm and 2.52 mm, corresponding to the approximate rostral and caudal limits of mPFC collected for RT-PCR analysis. (b) RefFinder comprehensive ranking (ranks are indicated under gene names on X-axis) based on the geometric mean of ranks obtained by four methods: (c) comparative Delta-Ct, (d) BestKeeper, (e) NormFinder, and (f) GeNorm. Since the GeNorm method calculates M-values based on gene expression ratios, the pair of genes with the highest stabilities share the same M-value.
(TIFF)

**S2 Fig. Stability values obtained by RefFinder online tool in the dorsal hippocampus (DH) of rats fed with medium-chain triglycerides (MCT).** MCT oil (2 ml/kg body weight) was given as a supplement to standard feed to adult (9 m.o., BW = 444 ± 69 g) animals by oral gavage daily for 4 weeks, while the control rats received equivolume water. (a) Coronal rat brain atlas diagrams (reproduced from Paxinos and Watson, 2007) showing DH (green) boundaries at Bregma -2.40 mm and -4.44 mm, corresponding to the approximate rostral and caudal limits of DH collected for RT-PCR analysis. (b) RefFinder comprehensive ranking (ranks are indicated under gene names on X-axis) based on the geometric mean of ranks obtained by four methods: (c) comparative Delta-Ct, (d) BestKeeper, € NormFinder, and (f)

GeNorm. Since the GeNorm method calculates M-values based on gene expression ratios, the pair of genes with the highest stabilities share the same M-value. Dashed line indicates GeNorm cut-off 0.5 M-Value: values above this line correspond to unstably expressed genes, which are invalid for RT-qPCR normalization. *–unstably expressed genes inappropriate for normalization according to the GeNorm cut-off 0.5 M-Value.
(TIFF)

**S3 Fig. Stability values obtained by RefFinder online tool in the ventral hippocampus (VH) of rats fed with medium-chain triglycerides (MCT).** MCT oil (2 ml/kg body weight) was given as a supplement to standard feed to adult (9 m.o., BW = 444 ± 69 g) animals by oral gavage daily for 4 weeks, while the control rats received equivolume water. (a) Coronal rat brain atlas diagrams (reproduced from Paxinos and Watson, 2007) showing VH (blue) boundaries at Bregma -4.44 mm and -5.28 mm, corresponding to the approximate rostral and caudal limits of VH collected for RT-PCR analysis. (b) RefFinder comprehensive ranking (ranks are indicated under gene names on X-axis) based on the geometric mean of ranks obtained by four methods: (c) comparative Delta-Ct, (d) BestKeeper, (e) NormFinder, and (f) GeNorm. Since the GeNorm method calculates M-values based on gene expression ratios, the pair of genes with the highest stabilities share the same M-value. Dashed line indicates GeNorm cut-off 0.5 M-Value: values above this line correspond to unstably expressed genes, which are invalid for RT-qPCR normalization. *–unstably expressed genes inappropriate for normalization according to the GeNorm cut-off 0.5 M-Value.
(TIFF)

**S4 Fig. Multiplex qPCR assay for *Actb*, *Gapdh*, and *B2m* genes demonstrate optimal efficiencies.** Efficiency was assessed by serial dilution method. Each standard curve was generated from a series of four-fold dilutions of pooled rat brain cDNA samples, collected in the main experiment.
(TIFF)

**S5 Fig. Multiplex qPCR assay for *Rpl13a*, *Sdha*, and *Ppia (CypA)* genes demonstrate optimal efficiencies.** Efficiency was assessed by serial dilution method. Each standard curve was generated from a series of four-fold dilutions of pooled rat brain cDNA samples, collected in the main experiment.
(TIFF)

**S6 Fig. Multiplex qPCR assay for *Hprt1*, *Pgk1*, and *Ywhaz* genes demonstrate optimal efficiencies.** Efficiency was assessed by serial dilution method. Each standard curve was generated from a series of four-fold dilutions of pooled rat brain cDNA samples, collected in the main experiment.
(TIFF)

## Author Contributions

**Conceptualization:** Alexander P. Schwarz.

**Data curation:** Alexander P. Schwarz, Alexander N. Trofimov.

**Formal analysis:** Alexander P. Schwarz, Alexander N. Trofimov.

**Funding acquisition:** Alexander P. Schwarz, Ksenia P. Shcherbakova, Alexander N. Trofimov.

**Investigation:** Veronika A. Nikitina, Darya U. Krytskaya, Alexander N. Trofimov.

**Methodology:** Veronika A. Nikitina, Darya U. Krytskaya, Ksenia P. Shcherbakova, Alexander N. Trofimov.

**Project administration:** Alexander N. Trofimov.

**Supervision:** Alexander N. Trofimov.

**Visualization:** Ksenia P. Shcherbakova, Alexander N. Trofimov.

**Writing – original draft:** Alexander P. Schwarz.

**Writing – review & editing:** Veronika A. Nikitina, Darya U. Krytskaya, Ksenia P. Shcherbakova, Alexander N. Trofimov.

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
