## [Decision Letter · Decision Letter 0]

14 Jul 2022

PONE-D-22-02598

Reference gene selection within the rat brain under mild intermittent ketosis induced by supplementation with medium-chain triglycerides

PLOS ONE

Dear Dr. Schwarz,

Thank you for submitting your manuscript to PLOS ONE. After careful consideration, we feel that it has merit but does not fully meet PLOS ONE’s publication criteria as it currently stands. Therefore, we invite you to submit a revised version of the manuscript that addresses the points raised during the review process.

ACADEMIC EDITOR:- please do follow directions of reviewers to improve your manuscript 

We look forward to receiving your revised manuscript.

Kind regards,

Prof. Dragan Hrncic, MD, PhD

Academic Editor

PLOS ONE

“The authors declare no conflict of interest.”

Reviewers' comments:

Reviewer's Responses to Questions

**Comments to the Author**

1. Is the manuscript technically sound, and do the data support the conclusions?

Reviewer #1: Yes

Reviewer #2: Yes

2. Has the statistical analysis been performed appropriately and rigorously? 

Reviewer #1: Yes

Reviewer #2: Yes

3. Have the authors made all data underlying the findings in their manuscript fully available?

Reviewer #1: Yes

Reviewer #2: Yes

4. Is the manuscript presented in an intelligible fashion and written in standard English?

Reviewer #1: Yes

Reviewer #2: Yes

5. Review Comments to the Author

Reviewer #1: In the present manuscript PONE-D-22-02598 authors investigated the effects of expression stability of 9 housekeeping genes widely used as reference and selected suitable reference genes for RT-qPCR analysis within the brain structures in rat model of mild ketosis. They used contemporary methodology and generated a large number of interesting results with potential interest to genetics-oriented readers. The manuscript should be improved for clarity according to provided comments and directions.

Minor points

Title

- Title should be changed to better reflect a full content of the manuscript.

- Maybe, one form of the title could be:” Reference gene expression stability within the rat brain under mild intermittent ketosis induced by supplementation with medium-chain triglycerides”

General

- English should be checked with corrections of some typos.

Abstract

- Please, introduce all abbreviations in appropriate places.

- Avoid starting this section with an abbreviation on the first place of the sentence.

- Address the impact of model of mild ketosis induced by medium-chain triglyceride supplementation and its relationship to genes expression.

Introduction

- Add a paragraph and explain the importance of selected genes, and describe some other studies which investigated the similar problematic.

- At the end of this section should be placed the aim of this study.

Material and methods

- In Animals and study design subsection, please add the number of ethical permission.

- Also, your manuscript will benefit by adding of one new figure with experimental timeline, which should be placed in Animals and study design subsection.

Results

- In Figures 3-5, for more clarity, colors could be additionally added, but it is not mandatory

Discussion

- Some typos should be corrected.

Reviewer #2: Positive comments

1. The paper is well written

2. The rationale behind executing the project is sound

3. The selection of initial reference genes for testing was based on the role of different genes in cell functions and also based on available literature

Revision Needed

1. Authors need to explain the PCR efficiency of the primers in the manuscript

2. Why were the authors using 2 mL/kg daily? Please give some supportive references.

3. *Line no. : 154: "RefFinder® online tool (https://www.heartcure.com.au/reffinder/)". Delete the website address as it was mentioned once in line 129.

4. In general, the authors should be writing medium-chain triglycerides (MCT) as either ‘MCT’ or ‘MCT oil’ in whole article.

5. Though gene expression was normalized with the three reference genes selected, this data still needs to be validated for gene expression with the bottom genes. This further proves the validity of the reference genes selected (Sahu., et al 2018)

6. PLOS authors have the option to publish the peer review history of their article (what does this mean?). If published, this will include your full peer review and any attached files.

Reviewer #1: **Yes: **Nikola Šutulović

Reviewer #2: No

---

## [Author Response · Author response to Decision Letter 0]

3 Aug 2022

Prof. Dragan Hrncic, MD, PhD

Academic Editor

PLOS ONE

Dear Dr. Hrncic,

Thank you and the reviewers for working on our manuscript and all the valuable comments and suggestions. We have prepared a revised version of the article with the changes marked. Briefly, we have corrected some typos throughout the manuscript and changed the title according to the reviewer's suggestion. 

Please find the detailed answers to the reviewers’ comments below.

Reviewer 1

«Title

- Title should be changed to better reflect a full content of the manuscript.

- Maybe, one form of the title could be:” Reference gene expression stability within the rat brain under mild intermittent ketosis induced by supplementation with medium-chain triglycerides”».

We changed the Manuscript title according to the reviewer’s suggestion.

«Abstract

- Please, introduce all abbreviations in appropriate places.

- Avoid starting this section with an abbreviation on the first place of the sentence.

- Address the impact of model of mild ketosis induced by medium-chain triglyceride supplementation and its relationship to genes expression.»

We edited the abstract to mention the full name of the method and added abbreviations’ disambiguation whereby needed (e.g. MCT, RT-qPCR). We also added two sentences describing our model of mild ketosis: this approach allows to reproduce certain neuroprotective effects of the classical ketogenic diet while avoiding its adverse effects. Ketogenic treatment targets multiple metabolic pathways, which may affect the reference gene expression. 

«Introduction

- Add a paragraph and explain the importance of selected genes, and describe some other studies which investigated the similar problematic.

- At the end of this section should be placed the aim of this study.»

We rearranged this section, moving the paragraph about the aim of the study to the end of the section. We also added information about the functional properties of the analyzed gene products.

«Material and methods

- In Animals and study design subsection, please add the number of ethical permission.

- Also, your manuscript will benefit by adding of one new figure with experimental timeline, which should be placed in Animals and study design subsection.»

We added the permission number and created a figure with the experimental timeline (Figure 1 in the revised version).

«Results

- In Figures 3-5, for more clarity, colors could be additionally added, but it is not mandatory»

We made colored versions of the graphs (figures 4-6 in the revised manuscript).

Reviewer 2.

1. «Authors need to explain the PCR efficiency of the primers in the manuscript».

We added the information about the range of measured PCR efficiencies to the Method section. The full information is presented in the Supplementary figures S4-S6.

2. «Why were the authors using 2 mL/kg daily? Please give some supportive references.».

We used this dosage to mimic typical low doses used in human studies. We added the supporting references in the Materials and Methods section.

3. «*Line no. : 154: "RefFinder® online tool (https://www.heartcure.com.au/reffinder/)". Delete the website address as it was mentioned once in line 129.»

We deleted the repeated mention of the URL.

4. In general, the authors should be writing medium-chain triglycerides (MCT) as either ‘MCT’ or ‘MCT oil’ in whole article.

We chose MCT for the whole article in the revised version.

5. Though gene expression was normalized with the three reference genes selected, this data still needs to be validated for gene expression with the bottom genes. This further proves the validity of the reference genes selected (Sahu., et al 2018)

We would like to thank Reviewer 2 for this suggestion, but we decided not to change our article in this way. Of course, the calculations for the genes of interest to the unstable genes could be used as a vivid illustration of the biases resulting from incorrect normalization. However, it would be a good approach if we were working with «marker» genes that moderately changed under the analyzed experimental conditions. In our case, we work with a new model, which is not well described in rats. So, we feel that the calculation of custom gene of interest expression normalized to different references would make our article more complicated and would not clarify the results.

We are looking forward for your review of the revised manuscript.

Alexander Schwarz

Ph.D., Research Fellow

Laboratory of Molecular Mechanisms of Neuronal Interactions

I.M. Sechenov Institute of Evolutionary Physiology and Biochemistry

St. Petersburg, Russia

Aleksandr.Pavlovich.Schwarz@gmail.com

---

## [Editor Report · Decision Letter 1]

5 Aug 2022

Reference gene expression stability within the rat brain under mild intermittent ketosis induced by supplementation with medium-chain triglycerides

PONE-D-22-02598R1

Dear Dr. Schwarz,

We’re pleased to inform you that your manuscript has been judged scientifically suitable for publication and will be formally accepted for publication once it meets all outstanding technical requirements.

Kind regards,

Prof. Dragan Hrncic, MD, PhD 

Academic Editor

PLOS ONE
---

## [Editor Report · Acceptance letter]

26 Sep 2022

PONE-D-22-02598R1 

Reference gene expression stability within the rat brain under mild intermittent ketosis induced by supplementation with medium-chain triglycerides 

Dear Dr. Schwarz:

I'm pleased to inform you that your manuscript has been deemed suitable for publication in PLOS ONE. Congratulations! Your manuscript is now with our production department. 

Kind regards, 

on behalf of

Professor Dragan Hrncic 

Academic Editor

PLOS ONE